# Ability of the Right Ventricle to Serve as a Systemic Ventricle in Response to the Volume Overload at the Neonatal Stage

**DOI:** 10.3390/biology11121831

**Published:** 2022-12-15

**Authors:** Chunxia Zhou, Debao Li, Qing Cui, Qi Sun, Yuqing Hu, Yingying Xiao, Chuan Jiang, Lisheng Qiu, Haibo Zhang, Lincai Ye, Yanjun Sun

**Affiliations:** 1Department of Thoracic and Cardiovascular Surgery, Shanghai Children’s Medical Center, School of Medicine, Shanghai Jiao Tong University, Shanghai 200127, China; 2Department of Cardiology, Shanghai Children’s Medical Center, School of Medicine, Shanghai Jiao Tong University, Shanghai 200127, China; 3Shanghai Institute for Pediatric Congenital Heart Disease, Shanghai Children’s Medical Center, School of Medicine, Shanghai Jiao Tong University, Shanghai 200127, China; 4Institute of Pediatric Translational Medicine, Shanghai Children’s Medical Center, School of Medicine, Shanghai Jiao Tong University, Shanghai 200127, China

**Keywords:** hormone, hypoplastic left heart syndrome, long-term management, neonatal, volume overload

## Abstract

**Simple Summary:**

The right ventricle (RV) of children with hypoplastic left heart syndrome (HLHS), in which volume overload (VO) is inevitable, pumps blood into the systemic circulation. Understanding the molecular differences and their different responses to VO between the RV and left ventricle (LV) at the neonatal and highly plastic stages may improve the long-term management of children with HLHS. Using our newly developed neonatal ventricular VO model, we demonstrated that one of the major differences between a normal neonatal RV and LV is related to the insulin and thyroid hormone signaling pathways. In response to VO, the RV favored an arrhythmogenic right ventricular cardiomyopathy (ARVC) phenotypic change and the LV favored a reduction in microRNAs in cancer. Considering that a major cause of death in children with HLHS is arrhythmia, which is the hallmark of ARVC, the current study suggests that inhibiting ARVC may improve RV function. In addition, the current study also suggests that insulin, thyroid hormone, and cancer-associated microRNAs may be potential therapeutic targets that should be explored by basic science studies to improve the function of the RV to match that of the LV.

**Abstract:**

Background: In children with hypoplastic left heart syndrome (HLHS), volume overload (VO) is inevitable, and the right ventricle (RV) pumps blood into the systemic circulation. Understanding the molecular differences and their different responses to VO between the RV and left ventricle (LV) at the neonatal and highly plastic stages may improve the long-term management of children with HLHS. Methods and Results: A neonatal rat ventricular VO model was established by the creation of a fistula between the inferior vena cava and the abdominal aorta on postnatal day 1 (P1) and confirmed by echocardiographic and histopathological analyses. Transcriptomic analysis demonstrated that some of the major differences between a normal neonatal RV and LV were associated with the thyroid hormone and insulin signaling pathways. Under the influence of VO, the levels of insulin receptors and thyroid hormone receptors were significantly increased in the LV but decreased in the RV. The transcriptomic analysis also demonstrated that under the influence of VO, the top two common enriched pathways between the RV and LV were the insulin and thyroid hormone signaling pathways, whereas the RV-specific enriched pathways were primarily associated with lipid metabolism and arrhythmogenic right ventricular cardiomyopathy (ARVC); further, the LV-specific enriched pathways were primarily associated with nucleic acid metabolism and microRNAs in cancer. Conclusions: Insulin and thyroid hormones may play critical roles in the differences between a neonatal RV and LV as well as their common responses to VO. Regarding the isolated responses to VO, the RV favors an ARVC change and the LV favors a reduction in microRNAs in cancer. The current study suggests that insulin, thyroid hormone, and cancer-associated microRNAs are potential therapeutic targets that should be explored by basic science studies to improve the function of the RV to match that of the LV.

## 1. Introduction

Hypoplastic left heart syndrome (HLHS) is an uncommon and complex congenital heart disease (CHD) with an incidence of about 1–27 per 100,000, accounting for about 1.4% of CHDs [1,2,3]. Although the incidence of HLHS is low, it accounts for 23% and 15% of cardiac deaths within the first week and the first month of life, respectively [4,5]. The 15-year survival of children with HLHS is only 48%, with most of these deaths occurring during the first year of life [4,5]. HLHS is characterized by a hypoplastic left ventricle (LV), systemic outflow tract obstruction, and pumping of systemic blood by the right ventricle (RV) [1]. Therefore, understanding the molecular differences between the neonatal RV and LV may aid in enhancing RV function, which improves the survival rate and long-term quality of life of children with HLHS.

One of the mechanisms underlying the causes of HLHS is the lack of sufficient volume load on the LV during the fetal stage [6,7]. In the embryonic period, the RV bears more volume load and is thicker than the LV; however, at term, the ventricular free wall thickness of both the LV and RV increases in parallel to almost 3.5 mm [8,9,10]. Increasing evidence suggests that neonatal hearts have a high degree of plasticity [11,12]. Our previous studies demonstrated that even at the prepubertal stage, volume overload (VO) still contributed to RV and LV adaptation [13,14,15]. To the best of our knowledge, no previous study has evaluated how VO remodels the neonatal RV and LV. Before the first-stage palliation, children with HLHS have adequate communication at the atrial level, which is required for survival and leads to RV VO [16,17]. During the first stage of palliative surgery (Norwood procedure), atrial septectomy is performed to produce unlimited pulmonary venous flow into the RV, which further increases VO to the RV [18,19,20]. After the third stage of correction (Fontan procedure, which is typically performed at preschool age) is completed, the RV simultaneously actively supports the systemic circulation and passively supports the pulmonary circulation [1,21,22,23]. It is unclear whether the function of the RV can be improved to match that of the LV in response to VO. Considering that most deaths due to HLHS occur within the first year of life [1,2,3], identifying the molecular differences between neonatal RV and LV, as well as their responses to VO, may help to improve the survival rate and the long-term performance of the RV in children with HLHS.

Previous studies have demonstrated that at the neonatal stage (from postnatal day 1 [P1] to P7), rodent cardiomyocytes (CMs) are immature, characterized by disorganized sarcomere arrangement, glycolysis metabolism, and a lack of T-tubules [11,24]. At the prepubertal stage (from P7 to P21), rodent CMs begin to mature, and by P21, the CMs are fully mature, characterized by a highly arranged sarcomere, lipid oxidative phosphorylation, and fully developed T-tubules [11,24]. Previously, we created a prepubertal ventricular VO model, showing that VO changed the prepubertal maturation track of both RV and LV and highlighting the higher degree of plasticity of prepubertal ventricles than that of adult ventricles [13,14,15]. In the current study, we constructed a neonatal and ventricular VO model on P1, as in our previous publication [25]. We selected the LV- and RV-free walls at P7 for analysis and compared them to demonstrate the molecular differences between neonatal RV and LV and to determine the differences in their responses to VO.

## 2. Materials and Methods

### 2.1. Animals and Surgery

Sprague–Dawley neonatal rats underwent either fistula surgery or sham operation at P1, as previously described, with associated video recordings to demonstrate the surgical process [25]. Briefly, the neonates were pre-anesthetized with 5% isoflurane and administered general anesthesia with ice cooling. The abdominal aorta (AA) and inferior vena cava (IVC) were exposed by midline laparotomy. Then, a fistula was created using a needle (diameter, 0.06 mm) puncture, directed from the AA to the IVC. A 2 min hemostatic compression was carried out after the puncture. Then, the abdominal wall was closed using local lidocaine treatment for pain relief. The neonatal rats were warmed on a heating plate at 37 °C for 2 min and then returned to their mother.

### 2.2. Abdominal Ultrasonography

A Vevo 2100 imaging system (Visual Sonics, Toronto, Ontario, Canada) and a pulse-wave mode were used to analyze the AA and IVC fistula (AVF), as reported previously [13,14,15].

### 2.3. Echocardiography

The creation of VO in AVF rats was confirmed by echocardiography using a Vevo 2100 imaging system (Visual Sonics, Toronto, Ontario, Canada). Moreover, the pulmonary artery (PA) velocity time integral (VTI), PA velocity, aortic valve (AoV) VTI, and AoV velocity were analyzed according to previous reports [13,14,15].

### 2.4. Histopathological Analysis

To evaluate the morphological changes in the ventricle caused by VO, the hearts of six rats per group were randomly selected to be stained with hematoxylin and eosin (H&E) using an H&E kit according to the manufacturer’s instructions.

### 2.5. Oil Red Stain

Oil red staining was performed using an Oil Red Stain Kit (ab150678; Abcam, Shanghai, China), as instructed by the manufacturer. In brief, the tissue sections were incubated in propylene glycol for 2 min, oil red O solution for 6 min, 85% propylene glycol for 1 min, and finally hematoxylin for 2 min.

### 2.6. Western Blotting Analysis

Proteins were extracted, separated, and transferred onto polyvinylidene fluoride membranes (Merck, Millipore, Billerica, MA, USA). The membranes were then blocked in 5% non-fat milk for 1 h at room temperature. After washing, the membranes were incubated with primary antibodies overnight at 4 °C. After washing, the membranes were incubated with secondary antibodies at room temperature for 1 h, and the blots were detected with a Bio-Rad ChemiDoc Imaging System (Bio-Rad, Hercules, CA, USA).

### 2.7. Library Construction and Sequencing

RV- and LV-free walls were selected for RNA-seq analysis. For RNA extraction, a PureLink RNA Micro Scale Kit was used. The NEBNext Ultra RNA Library Prep Kit for Illumina (NEB, USA) was utilized to generate sequencing libraries. Briefly, mRNA was first purified with poly-T oligo-attached magnetic beads. Then, we performed fragmentation with First Strand Synthesis Reaction Buffer. First-strand cDNA was synthesized using RNase H and random hexamer primers, and second-strand cDNA was synthesized with DNA polymerase I and RNase H. The cDNA fragments were first adenylated and then ligated with NEB Next Adaptors. An AMPure XP system (Beckman Coulter, Beverly, MA, USA) was used to purify library fragments that were preferentially 370–420 bp in length. Then, we performed PCR with Index (X) Primer, Universal PCR primers, and Phusion High-Fidelity DNA polymerase. Subsequently, we purified the PCR products with an AMPure XP system and assessed the library quality with the Agilent Bioanalyzer 2100 system. The samples were finally clustered with a TruSeq PE Cluster Kit v3-cBot-HS (Illumina) and sequenced on an Illumina Novaseq platform.

### 2.8. Quality and Quantification of Gene Expression Levels

To generate clean data (clean reads), reads containing adapters or poly-N or reads with low quality were removed from the raw data, and the downstream analyses were based on the clean data. Hisat2 v2.0.5 was used to establish the index of the reference genome and to align the paired-end clean reads to the reference genome. The featureCounts v1.5.0-p3 was employed to count the number of reads mapped to each gene. Gene expression levels were expressed as fragments per kilobase of transcript sequence per million base pairs sequenced (FPKM).

### 2.9. Analysis of Differentially Expressed Genes

The DESeq2 R package (1.16.1) was used to determine differentially expressed genes (DEGs). Benjamini and Hochberg’s approach was applied to adjust the resulting P values. Genes with an adjusted *p* value of <0.05 and fold change > 1.3 were considered to be differentially expressed. The clusterProfiler R package was used to analyze Gene Ontology (GO) and KEGG pathway enrichment. Terms with corrected *p* values of <0.05 were considered to be significantly enriched.

### 2.10. Statistical Analysis

Continuous data are expressed as means ± standard deviation. When the data were normally distributed, as shown by the Shapiro–Wilk test, Student’s *t*-test was used to test the differences between groups; otherwise, the data were analyzed with the rank-sum test. *p* < 0.05 was considered to indicate a significant difference. Statistical analyses were conducted using SAS software ver. 11.0 (SAS Institute Inc., Cary, NC, USA).

## 3. Results

### 3.1. Establishment of Ventricular VO via AVF Surgery

Similar to our previous publications [13,14,15], no pulsatile blood flow was shown in the IVC of the AVF rats (Figure 1A); however, pulsatile blood flow appeared in the AA (Figure 1B) and at the puncture point (PP). The mean peak velocity at the PP was 286.7 ± 17.2 (Figure 1C,D). The data demonstrated that a fistula was created between the AA and IVC.

To evaluate the RV VO, the PA velocity and PA velocity time integral (VTI) at P7 were detected. The PA velocities and PA VTIs in the VO groups significantly increased (Figure 2A–C) when compared to the sham group. Similarly, in the LV, the AoV velocities and AoV VTIs in the VO groups significantly increased when compared to the sham group (data not shown). H&E staining showed that the diameters of the RV and LV chambers were increased at the middle panel in the VO group (Figure 2D–F). These results suggested that VO was successfully created by AVF in both the RV and the LV, as previously reported [13,14,15].

### 3.2. Molecular Differences between Neonatal LV and RV

To explore how VO contributes differently to RV and LV, we compared the transcriptome between normal neonatal RV and LV by RNA-seq. As shown in Figure 3A, there were 1806 differentially expressed genes (DEGs) between neonatal LVs and RVs, of which 1095 were downregulated and 711 were upregulated. When these genes were clustered, a heatmap showed that the DEGs of the individual RVs were similar to each other but differed significantly from those of the LVs (Figure 3B), suggesting good reproducibility of the study. Enrichment analysis of the DEGs between normal neonatal LVs and RVs showed that among the top 10 enriched terms for biological process (BP), 6 were associated with signal transduction, whereas calcium ion binding and protein kinase activity were among the top 10 enriched terms for molecular function (MF) (Figure 3C and Appendix A). Because intracellular signal transduction depends on calcium ion binding and protein kinase activity [26,27], the above results suggested that one of the major molecular differences between the normal neonatal RV and LV was related to intracellular signal transduction.

To further understand the regulation of molecular differences between the LV and RV by intracellular signaling, a pathway enrichment analysis was applied. The results showed that the top four enriched pathways were the thyroid hormone signaling pathway, insulin resistance, cGMP-PKG signaling pathway, and insulin signaling pathway (Figure 3D and Appendix A). 

The above-mentioned results suggest that one of the major differences between neonatal LV and RV was in the thyroid hormone and insulin signaling pathways.

To verify the results demonstrated by RNA-seq, we detected the thyroid hormone and insulin receptors. The results showed that the levels of thyroid hormone and insulin receptors were significantly lower in the LV than in the RV (Figure 4A–G). In addition, under the condition of VO, thyroid hormone receptor (Thra) and insulin receptor (Insr) levels were upregulated in the LV but were downregulated in the RV (Figure 4A–G).

### 3.3. Neonatal LV Was more Transcriptionally Active at Baseline in Response to VO Than the RV

As shown in Figure 5A, under the influence of VO, there were 922 DEGs in the RV comparison, while there were 3408 DEGs in the LV comparison, suggesting that VO produced almost a 4-fold increase in DEGs in the LV in comparison to the RV. A Venn diagram showed that the LV comparison shared 530 DEGs with the RV comparison; 2878 DEGs were expressed only in the LV comparison, and 392 DEGs were only expressed in the RV comparison (Figure 5B). As shown in Figure 5C, when the DEGs were clustered in the RV or LV comparison, a heatmap showed that the individuals were similar to each other in the same group but differed significantly from those in the other group, suggesting good reproducibility of the study. The results suggested that the neonatal LV was more transcriptionally active at baseline in response to VO than the RV.

### 3.4. Common Responses between Neonatal LV and RV under the Influence of VO

The aforementioned DEGs were subjected to an enrichment analysis. In the top 150 enriched terms (50 terms each for biological process, cellular component, and molecular function) and 30 pathway terms, LV and RV comparisons shared 5 biological process terms (10%) (Appendix A), 13 cellular component terms (26%) (Appendix A), 16 molecular function terms (32%) (Figure 6A), and 14 pathway terms (36.7%) (Figure 6B). The scores of enriched terms in the LV comparison were higher than those in the RV comparison, except for the terms of insulin resistance and thermogenesis in pathways (Figure 6B), thereby confirming that the neonatal LV was more transcriptionally active in response to VO than the RV. Although the normal neonatal LV and RV differed in terms of the thyroid hormone and insulin signaling pathways (Figure 3D and Appendix A), the thyroid hormone and insulin signaling pathway were the top two enriched terms in both the LV and the RV under the influence of VO (Figure 6B), suggesting that thyroid hormone and insulin play a critical role in regulating their common responses to VO.

### 3.5. Different Responses of Neonatal LV and RV under the Influence of VO

The aforementioned DEGs were subjected to enrichment analysis which showed the top 30 RV- or LV-specific enriched terms in response to VO (Figure 7). In the RV, VO caused BP enrichment that was associated with protein and lipid metabolism (Figure 7A), while in the LV, it caused BP enrichment that was associated with nucleoside metabolism (Figure 7B). Further, VO caused CC enrichment that was associated with protein and lipid synthesis sites in the RV, while it was associated with vesicles in the LV (Appendix A). Moreover, VO caused MF enrichment that was associated with protein and lipid-metabolic enzyme activities in the RV, while it was associated with nucleoside metabolic regulation in the LV (Appendix A).

In addition, the top three pathways enriched by VO in the RV were arrhythmogenic right ventricular cardiomyopathy (ARVC), the adipocytokine signaling pathway, and the PPAR signaling pathway (Figure 8A), whereas in the LV they were microRNAs in cancer, the longevity regulating pathway, and the citrate cycle (TCA cycle) (Figure 8B). ARVC, the adipocytokine signaling pathway, and the PPAR signaling pathway are closely associated with the regulation of protein and lipid metabolism [28,29,30]. The microRNAs in cancer and longevity-regulating pathways are closely associated with nucleic acid metabolism [31,32]. These results suggested that under the influence of VO, the RV favors the regulation of protein and lipid metabolism, while the LV favors the regulation of nucleic acid metabolism.

To confirm the above results, we detected lipid metabolism in the RV and cancer-associated microRNAs in the LV. Although the adipocytes (a marker of ARVC) were not increased in the RV under the influence of VO (Figure 9A), the lipid metabolism enzyme long-chain acyl-CoA synthetase 1 (ACSL1) was significantly increased in the RV (Figure 9B,C). Moreover, cancer-associated microRNAs were downregulated in the LV (Figure 9D).

## 4. Discussion

The mortality of children with HLHS is the highest between stage 1 and stage 2 of the operation, during which VO is inevitable and has been associated with arrhythmias and RV failure [1,2]. Previous studies have focused mainly on the hemodynamic changes rather than on the molecular changes at this stage [3,4]. To the best of our knowledge, the current study is the first to demonstrate the transcriptomic differences between the neonatal RV and LV as well as their responses to VO, thereby providing useful information to improve the function of the RV to support the systemic circulation and to reduce the rate of arrhythmias and RV failure.

Our results demonstrated that the insulin and thyroid hormone signaling pathways were the two of the most enriched pathways regulating the differences between the RV and LV as well as their common responses to VO, suggesting that these two hormone pathways may be used to improve RV function. Previous studies demonstrated that insulin promotes the use of glucose as the main cardiac energy substrate, resulting in reduced myocardial O2 consumption and increased cardiac efficiency [33,34]. As a result, insulin resistance contributes to VO-induced adult heart failure and prepubertal cardiac metabolic maladaptation [35,36]. To our knowledge, the present study is the first to reveal the role of insulin in VO-stimulated neonatal RV and LV responses. Thyroid hormone leads to increased expression of the fast isoform of the myosin heavy chain and sarcoplasmic reticulum calcium-activated ATPase and increased beta-1-adrenergic receptor density, enhancing both systolic and diastolic LV function [37,38]. In addition, recent studies have demonstrated that thyroid hormone controls cardiac plasticity, including cardiac regeneration [39]. Based on the results of the current and previous studies, we suggested that insulin and thyroid hormones may be used to support neonatal RV function in response to VO. Moreover, our previous studies demonstrated that VO induced an immune response of the neonatal RV and prepubertal LV [13,15,25], whereas the current study showed that VO induced an immune response of both RV and LV (Appendix A). In addition, adult hearts also respond to VO with an immune response [40,41]. This confirmed our previous results and highlighted the role of immune response in ventricular maladaptation. However, fibrosis and angiogenesis were the two hallmarks of adult hearts in response to VO [40,41]. However, neonatal hearts exposed to VO, either in the RV or LV, did not show any enriched BP, CC, MF, or pathway terms associated with fibrosis and angiogenesis (Figure 6). Additionally, the enriched terms associated with insulin and thyroid hormones in neonatal hearts did not appear in adult hearts [40,41].

Another finding of the present study is that the responses between the neonatal RV and LV under the influence of VO are different. The hallmarks of ARVC are arrhythmia, adipocytes, and an imbalance of calcium homeostasis [28]. In the RV, the BP enriched by VO was mainly associated with protein and lipid metabolism (Figure 7A), whereas the CC was mainly associated with protein and lipid synthesis sites (Appendix A), and the MF was mainly associated with protein- and lipid-metabolic enzyme activities (Appendix A). Similarly, the pathways enriched by VO in the RV were mainly associated with ARVC and the adipocytokine signaling pathway (Figure 8A). These results indicated that under the influence of VO, neonatal RV developed an AVRC-like phenotype, which partly explains why some children with HLHS died of arrhythmia. In contrast, the neonatal LV uses cancer-associated microRNAs to adapt to VO (Figure 8B). Thus, the function of the RV may be improved to match that of the LV using ion channel blockers and cancer-associated microRNAs, which are potential therapeutic targets that should be explored by basic science studies.

An important limitation of the current study is that we did not use a transgenic mouse to evaluate the role of insulin, thyroid hormone, ion channel blockers, and cancer-associated microRNAs in RV performance. Further studies are required to extend our evaluation of the effects of insulin, thyroid hormone, ion channel blockers, and cancer-associated microRNAs on RV performance. Another major concern is that the model cannot exactly mimic HLHS and, to some extent, is much more similar to left-to-right shunt. However, AVF produces VO not only in the RV but also in the LV and has been extensively used to study the effect of VO on both the RV and LV [13,42,43]. Thus, this model provides an alternative method to study how VO affects LV and RV, especially at the neonatal stage, at which no other ventricular VO models have been developed [13,14,15]. The third major concern is whether the LV is more capable than the RV in maintaining persistent function under VO. Most of our knowledge concerning the different responses to VO between the LV and RV was obtained from adult animals. For example, based on the study of adult animals, we know that the LV was more sensitive to VO than the RV, whereas the RV was more sensitive to pressure overload than the LV [42]. One may expect that the LV would not demonstrate higher capability than the RV to maintain persistent function under VO. However, neonatal hearts are quite different from adult hearts in terms of their molecular structure, metabolism, and electrophysiology [12], and we should not apply our knowledge obtained from adult hearts to neonatal hearts. For example, one cause of HLHS is the lack of volume load at the fetal stage [6]. However, one would not expect volume load to contribute to adult LV development. Moreover, the adult LV responds to pressure overload with dysfunction, while the neonatal LV responds to the same stimulus with proliferation and a pure adaptive response with preserved cardiac function [44,45]. When considering HLHS, the RV replaces the LV as a systemic ventricle. This means that the RV bears a higher load (volume load, pressure load, or both) than under normal circumstances. The current study aimed to shed light on the fact that the RV may be rebuilt and may function better than the LV from the aspect of volume load. In addition, the RV has two layers and two types of fibers (clockwise spiral fibers and longitudinal fibers), whereas the LV has three layers and three types of fibers (clockwise spiral fibers, anti-clockwise spiral fibers, and longitudinal fibers) (Appendix A), rendering the LV more resistant to dilatation than the RV [42]. Thus, with increasing volume load, the LV is more prone to dysfunction than the RV at the adult stage [42]. Whether a similar phenomenon exists at the neonatal stage was largely unknown and was a part of the question the study tried to answer. As shown in Figure 5A, our results demonstrated that the LV was more sensitive to VO than the RV, producing 4 times more DEGs.

## 5. Conclusions

In summary, we found that insulin and thyroid hormones were involved in the common responses to VO for both the neonatal RV and the neonatal LV, suggesting that these hormones may be used to improve RV performance. We also demonstrated that under the influence of VO, the RV transformed to an AVRC-like phenotype, while the LV demonstrated reduced production of cancer-associated microRNAs, suggesting that ion channel blockers and cancer-associated microRNAs are potential therapeutic targets that should be explored by basic science studies.

## Figures and Tables

**Figure 1 biology-11-01831-f001:**
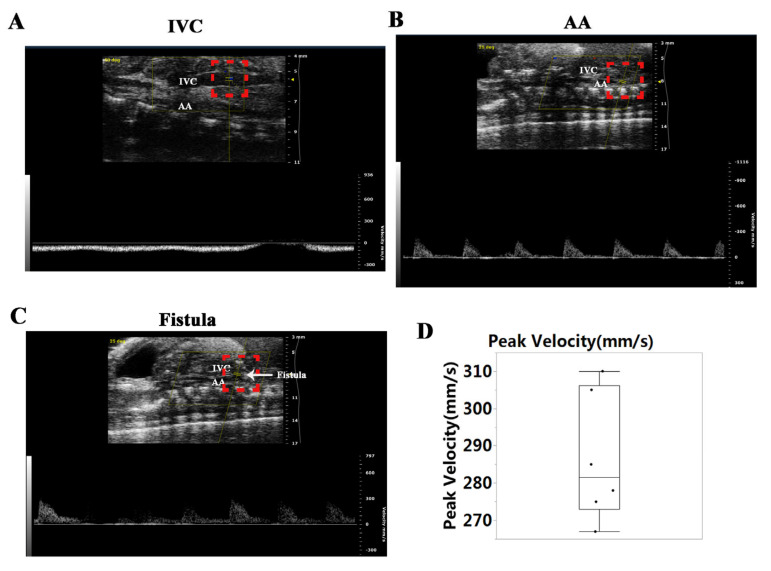
Establishment of the abdominal aorta (AA) and inferior vena cava (IVC) fistula (AVF). (**A**) IVC manifested no pulsatile blood flow. (**B**) There was pulsatile blood flow in the AA, with a peak blood flow velocity of 200 mm/s. (**C**) Representative image of the pulsating blood flow at the fistula, with a peak blood flow velocity of 300 mm/s. (**D**) Quantification of peak velocity at the fistula.

**Figure 2 biology-11-01831-f002:**
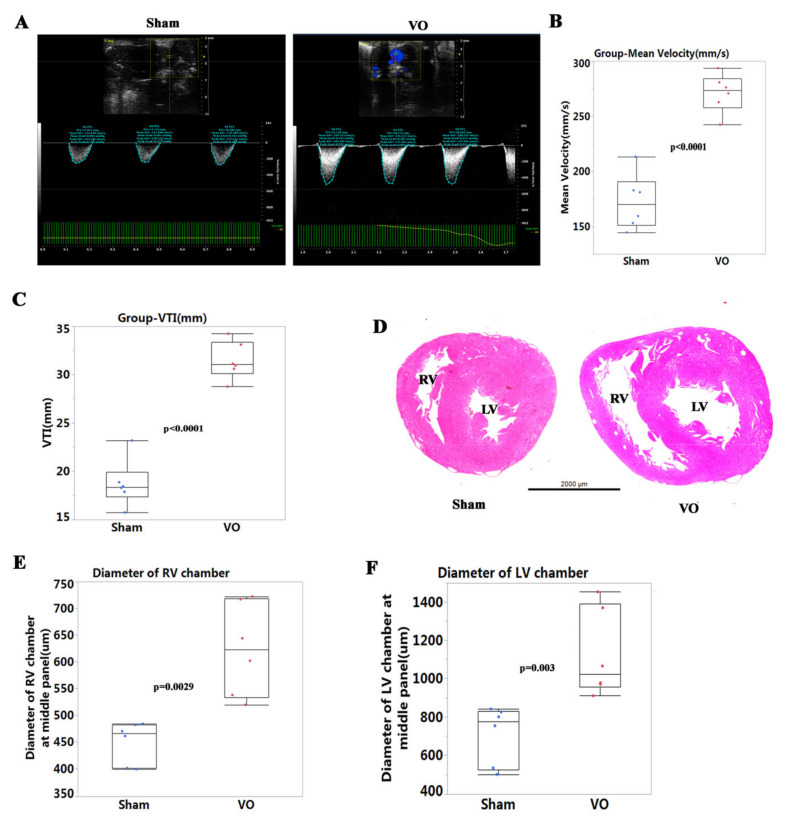
Verification of volume overload (VO) in the AVF group. (**A**) Representative echocardiogram of pulmonary artery (PA) velocity time integral (VTI) and velocity in the sham and VO groups. (**B**) Quantification of PA mean velocity in the sham and VO groups, *n* = 6. (**C**) Quantification of PA VTI in the sham and VO groups, *n* = 6. (**D**) Representative hematoxylin and eosin staining of hearts from the sham and VO groups. (**E**) Quantification of the diameter of the right ventricle (RV) chamber. (**F**) Quantification of the diameter of the left ventricle (LV) chamber.

**Figure 3 biology-11-01831-f003:**
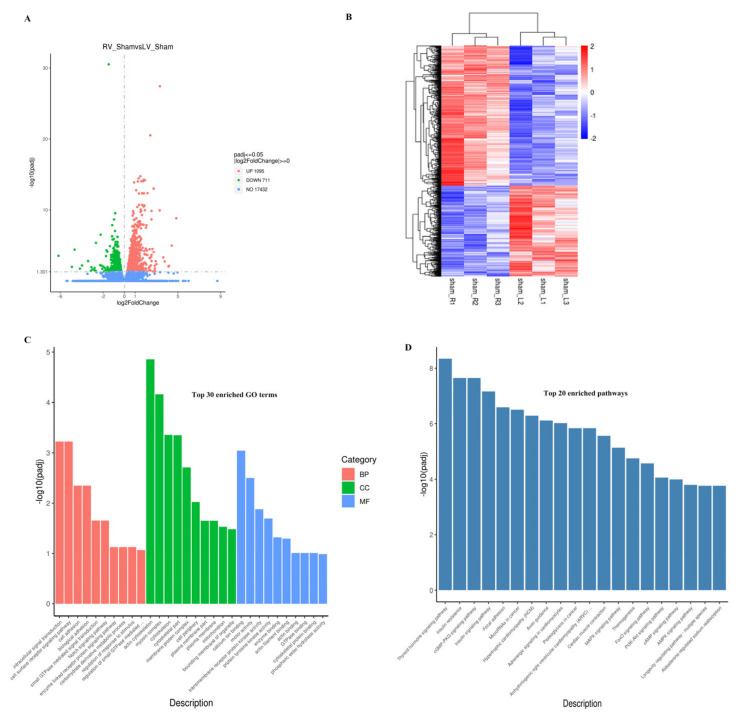
Molecular differences between neonatal right ventricle (RV) and left ventricle (LV). (**A**) Volcano map of the differentially expressed genes (DEGs) between neonatal LV and RV. There were 711 upregulated and 1095 downregulated genes in the normal LV compared to the normal RV. (**B**) Cluster analysis of the DEGs between neonatal LV and RV. The clusters of genes in each group were different from each other but were similar within the same group. (**C**) Histogram of the GO analysis. Based on the results of the GO enrichment analysis, we displayed the 10 most significant terms. The abscissa is the GO term, and the ordinate is the significance level for GO term enrichment. The higher the value, the more significant the results. The different colors represent three different GO subclasses: biological process (BP), cellular component (CC), and molecular function (MF). (**D**) Histogram of the 20 most significant KEGG pathways. The 20 most significant Kyoto Encyclopedia of Genes and Genomes (KEGG) pathways from the KEGG enrichment results are shown. The abscissa is the KEGG pathway and the ordinate is the significance level of pathway enrichment; the higher the value, the greater the significance.

**Figure 4 biology-11-01831-f004:**
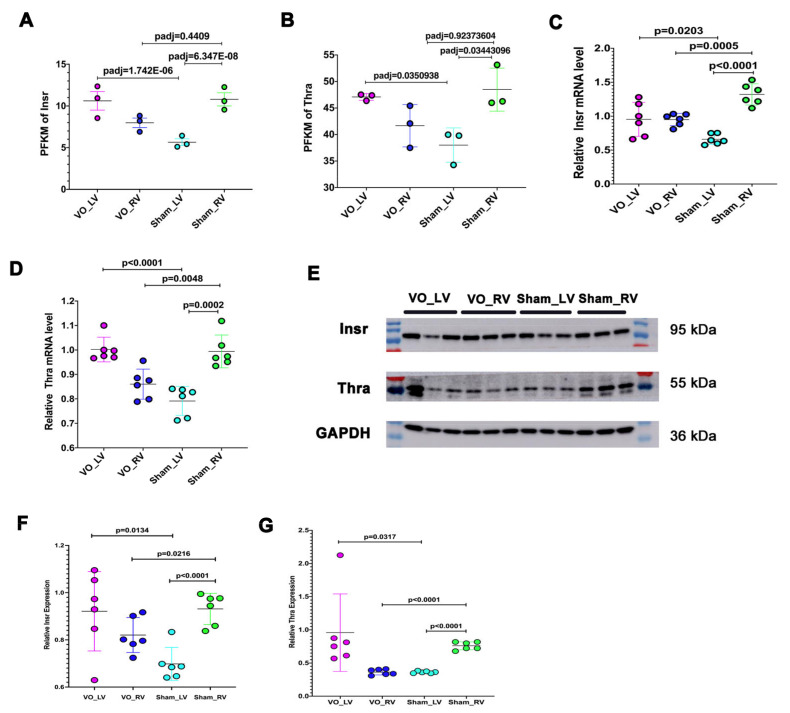
Expression of thyroid and insulin receptors in the left ventricle (LV) and right ventricle (RV). (**A**) FPKM of insulin receptor (Insr). (**B**) FPKM of thyroid receptor alpha (Thra). (**C**) The mRNA level of Insr. (**D**) The mRNA level of Thra. (**E**) Representative Insr and Thra blots. (**F**) Quantification of Insr blots. (**G**) Quantification of Thra blots. It is noted that Insr and Thra were significantly lower in the LV than in the RV, and when subjected to VO, Insr and Thra were upregulated in LV but downregulated in the RV.

**Figure 5 biology-11-01831-f005:**
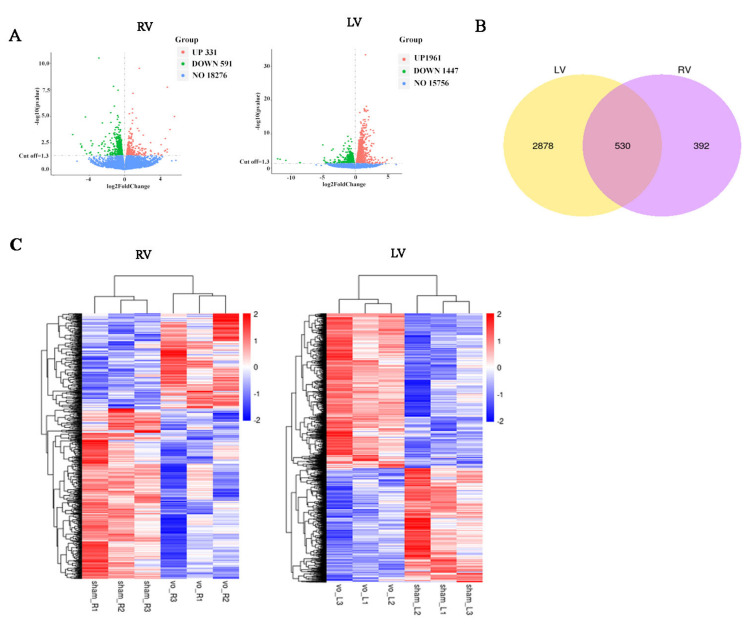
Left ventricle (LV) was more sensitive to VO than the right ventricle (RV). (**A**) Volcano map of the differentially expressed genes (DEGs) between VO and sham groups. There were 926 DEGs in the RV comparison and 3408 DEGs in the LV comparison. (**B**) Venn map of the LV comparison and RV comparison. The LV comparison shared common 530 DEGs with the RV comparison, 2878 DEGs were expressed only in the LV comparison, and 392 DEGs were only expressed in the RV comparison. (**C**) Cluster analysis of the DEGs between VO and sham groups. The clusters of genes in each group were quite different from each other but were similar within the same group.

**Figure 6 biology-11-01831-f006:**
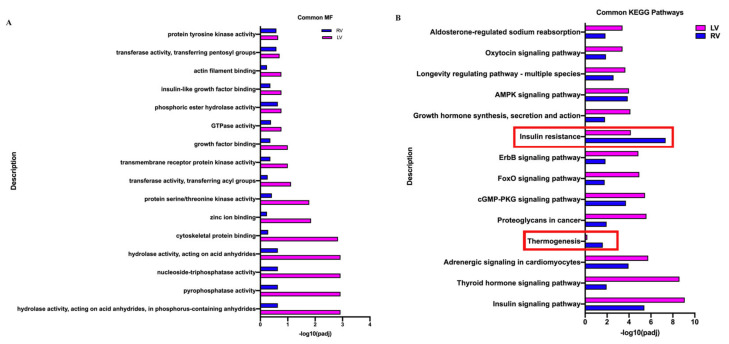
Common responses to volume overload (VO) between the neonatal left ventricle (LV) and right ventricle (RV). (**A**) Molecular function (MF) terms of GO analysis shared by LV and RV. (**B**) KEGG signaling pathways terms shared by LV and RV. The scores in terms of insulin resistance and thermogenesis(highlighted in red box) in the LV comparison were lower than in the RV comparison.

**Figure 7 biology-11-01831-f007:**
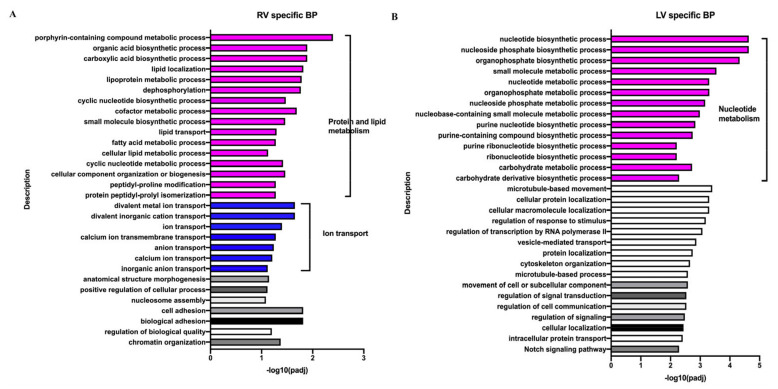
Different responses to volume overload (VO) between the neonatal left ventricle (LV) and right ventricle (RV). (**A**) Top 30 RV-unique biological process (BP) terms from GO analysis of differentially expressed genes (DEGs) between sham and VO groups. (**B**) Top 30 LV-unique biological process (BP) terms from GO analysis of DEGs between sham and VO groups.

**Figure 8 biology-11-01831-f008:**
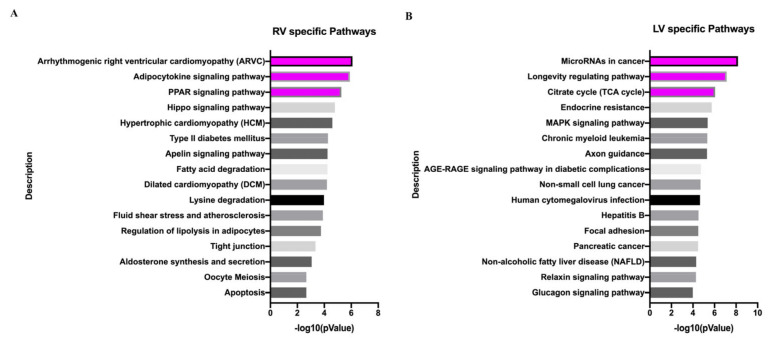
Different KEGG pathways between the neonatal left ventricle (LV) and right ventricle (RV) in response to volume overload (VO). (**A**) Top 16 RV-unique terms of KEGG pathway analysis of differentially expressed genes (DEGs) between sham and VO groups. (**B**) Top 16 LV-unique terms of KEGG pathway analysis of DEGs between sham and VO groups.

**Figure 9 biology-11-01831-f009:**
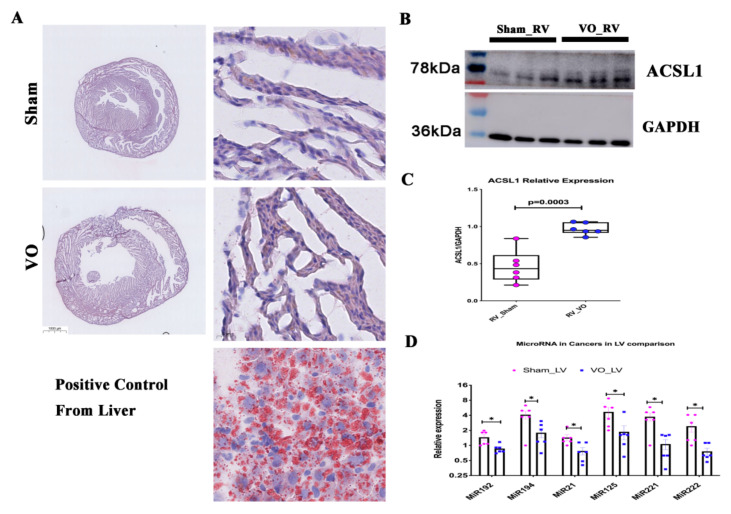
Verification of unique responses in the right ventricle (RV) or left ventricle (LV). (**A**) Representative image of Oil red staining. In the control, adipocytes were stained with oil red. (**B**) Representative ACSL1 blots. (**C**) Quantification of ACSL1 blots. (**D**) Relative expression of cancer-associated microRNAs. * *p* < 0.05.

## Data Availability

The RNA-seq data were deposited in the GEO database (https://www.ncbi.nlm.nih.gov/geo (accessed on 5 December 2022)) with accession number GSE186247.

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
