# Peer review of "Ability of the Right Ventricle to Serve as a Systemic Ventricle in Response to the Volume Overload at the Neonatal Stage"

_biology, 2022, doi:10.3390/biology11121831_

Round 1

Reviewer 1 Report

Thank you for the opportunity to review this study by Zhou et al. which analyzes the molecular signatures of a volume loaded right ventricle using an abdominal aorta – IVC fistula rat model. The authors have previously published on this topic and the study at hand is well written with novel findings. This review serves primarily as a grammatical and structural review. The authors should address the following points before this article is published.

1.     Simple Summary:

a.     The sentence “We found that the common response…” on line 26 can be removed as it is redundant

2.     Abstract:

a.     Revise “LV-specific enriched pathways were primarily associated nucleic…” to “LV-specific enriched pathways were primarily associated with nucleic…” on line 48

b.     Remove or clarify the sentence “The ARVC markers and cancer-associated…” on line 49

3.     Graphical Abstract:

a.     Add “VO”, “ARVC” to abbreviations list

b.     Ensure title of graphical abstract matches article title

4.     Introduction:

a.     Replace “but” with “and” on line 61

b.     Revise “leading to RV VO and is required for survival” on line 80, perhaps to the following: “Before the first-stage palliation, children with HLHS have communication at the atrial level, which is required for survival and also leads to RV VO”

c.     Revise “the RV compensates for the lack…” on line 84 to the following, “the RV simultaneously supports the systemic circulation actively and the pulmonary circulation passively”

d.     Revise “it is unclear whether…” on line 85 to the following “It is unclear whether the function of the RV can be improved to match that of the LV in response to VO”

e.     The statement “Considering that most HLHS children die within the first year of life…” is not true. Revise to “Considering that most deaths due to HLHS occur within the first year of life…”

5.     Results

a.     Revise the sentence for clarity “…these results suggested that the thyroid hormone and insulin signaling pathways are some of the major differences between neonatal LV and RV” on line 223

b.     Figure 3A resolution is poor, please recreate

c.     Figure 3, please define the abbreviation “KEGG”

d.     Please reword sentence on line 237, “To confirm the RNAseq results…”

e.     Figure 5, please consider increasing size of font on x-axis for 5C as it is not readable and perhaps relabeling clearly to “SHAM 1”, “VO 1”, etc.

6.     Discussion

a.     Do the authors really think it is feasible to use “ion channel blockers and cancer-associated microRNAs” for post-surgical treatment of HLHS knowing what we know from this study? Perhaps this should be revised to state that these are potential therapeutic targets that future basic science studies should explore. All we know is that these pathways are differentially expressed in the LV in response to volume overload, not necessarily that they are beneficial…

b.     “Persisted function” replace with “persistent function” on line 388

c.     Just a general comment, can we necessarily conclude from the fact that the LV expressed more differentially expressed genes in response to VO, that the LV is more “sensitive” to volume overload? Perhaps the LV is more transcriptionally active at baseline.

Reviewer 2 Report

The topic is important and relevant to the journal scope,the writing style is good ,figures are of very  good quality except figure 1 very bad in quality must be enlarged enough with good resolution...please check... all figure are of good presentation...the study is  somewhat novel with new presentation and expression of data...I have some recommendations;please address in the revised version...  

Title : please add the before voulme overload 

Abstract: good but the conclusion must be more informative please add details on the conclusion of abstract 

Keywords:must be arranged alphabetically with first capital letter 

Introduction: very good and informative  but authors  must say enough about this paragraph (line 90-93)

Materials:

Line 101 where is the certified ethical commitee approval number 

Line 121 there is no information about ultrasound setting device????

Line 125:what about study inclusion and exclusion criteria.

Line 167 in statistical analysis author did not state how data is checked first is data are normal and check by one of the normality tests or not??? Please answer this critical questions....

Results are good in presentation but need to me more summarized. ..

In the discussion part please pay attention of typing errors and grammer errors with much more information should be present in this section with other previous studies......

Reviewer 3 Report

The authors used neonatal rat to establish a ventricular volume overload (VO) model. Transcriptome sequencing was performed on the left and right ventricles, respectively, to understand the molecular differences between RV and left ventricle and their different responses to VO, and look forward to seeking long-term treatment methods that can potentially improve HLHS in children. However, this VO model is not representative of HLHS. This manuscript could be only used for reference as a bioinformatics article. The following comments are made for further consideration.

1.  In the paragraph of “Common responses between neonatal LV and RV under the influence of VO“, the authors need to analyze the up-regulated genes and down-regulated genes of left ventricle and right ventricle in detail, not just the differential genes.

2.  Authors found the important roles of insulin, thyroid, ion channel blockers, and cancer-associated microRNAs by RNA seq. So authors need to evaluate them in VO model.

3. There are other similar manuscripts about RNA seq of VO in heart(for example, doi: 10.3389/fcvm.2021.772336; DOI: 10.1161/JAHA.119.015574), why do similar models and methods lead to different results? Authors also need to analyze the common and different responses between neonatal and adult heart under the influence of VO in discussion.

Round 2

Reviewer 3 Report

I have no other question.